# Photo-Crosslinking Hydrogel Based on Porcine Small Intestinal Submucosa Decellularized Matrix/Fish Collagen/GelMA for Culturing Small Intestinal Organoids and Repairing Intestinal Defects

**DOI:** 10.3390/ijms26020663

**Published:** 2025-01-14

**Authors:** Zihao Jia, Ziwei Wang

**Affiliations:** Department of Gastrointestinal Surgery, The First Affiliated Hospital of Chongqing Medical University, Chongqing 400010, China; drjiazihao@sina.com

**Keywords:** photo-crosslinking hydrogel, porcine small intestinal submucosa decellularized matrix, fish collagen, small intestinal organoid, intestinal defect repair

## Abstract

Organoid technology, as an innovative approach in biomedicine, exhibits promising prospects in disease modeling, pharmaceutical screening, regenerative medicine, and oncology research. However, the use of tumor-derived Matrigel as the primary method for culturing organoids has significantly impeded the clinical translation of organoid technology due to concerns about potential risks, batch-to-batch instability, and high costs. To address these challenges, this study innovatively introduced a photo-crosslinkable hydrogel made from a porcine small intestinal submucosa decellularized matrix (SIS), fish collagen (FC), and methacrylate gelatin (GelMA). The cost-effective hydrogel demonstrated excellent biocompatibility, tunable mechanical properties, rapid gelation properties, and low immunogenicity. Importantly, the proliferation and differentiation capacities of small intestinal organoids cultured in hydrogel were comparable to those in Matrigel, with no significant disparity observed. Furthermore, after one week of transplantation in nude mice, the hydrogel–organoid complex exhibited sustained structural and functional stability while preserving the differentiation characteristics of small intestinal organoids. Our study also demonstrated the effective potential of FC/SIS/GelMA hydrogel in accelerating the repair process of small intestinal defects, reducing the area of scar formation, and promoting the regeneration of both intestinal villi and smooth muscle tissue. In summary, this study presents a novel protocol for culturing small intestinal organoids, offering potential implications for future clinical applications and serving as an experimental foundation for the development of tissue-engineered intestines based on small intestinal organoids.

## 1. Introduction

The technology of small intestinal organoids provides a novel approach for exploring gut development [1,2], disease modeling [3,4,5], personalized medicine [6,7,8], and small bowel regeneration [9,10,11]. The clinical translation of small intestinal organoids remains, however, significantly limited by their reliance on Matrigel for culturing, owing to several factors. These challenges stem from the following factors: (1) Matrigel derived from tumors carries a potential risk of tumorigenicity. (2) Substantial batch-to-batch variations. (3) The operational costs are prohibitively high [12,13,14,15]. To overcome the aforementioned issues, extensive research has been conducted in recent years, leading to the identification of various biomaterials capable of facilitating the culture of small intestinal organoids. Promising examples include plant-based nanocellulose hydrogel [16], non-adhesive alginate hydrogels [17], and decellularized porcine small intestine mucosa/submucosa hydrogel [18]. While these materials demonstrate potential, limited reports exist regarding their clinical translation. Motivated by this gap, we envisioned and fabricated an economical hydrogel capable of both culturing small intestinal organoids and promoting the repair of small intestinal defects.

The advantages of fish collagen (FC) include its cost-effectiveness, low immunogenicity, excellent biocompatibility and bio-degradability, the presence of multifunctional groups, high water absorption capacity, and ease of processing [19]. As a result, it has been widely utilized in regenerative medicine applications such as wound healing, anti-aging skin treatments, corneal tissue engineering, vascular tissue regeneration, and bone and cartilage regeneration [20,21,22]. Briefly, it has the following advantages [23]: (1) Diverse sources and cost-effectiveness. (2) The absence of religious constraints. (3) The absence of zoonotic risks. These attributes have made fish collagen highly valuable across the food, pharmaceutical, and biomedical industries [21]. A wealth of scientific literature has substantiated the beneficial impact of fish collagen on tissue regeneration, specifically in the field of wound healing [24]. Additionally, fish collagen has demonstrated anti-inflammatory properties through the inhibition of TNFα, thereby facilitating the repair of colitis and intestinal ulceration [25]. Despite these promising attributes, the potential of fish collagen for cultivating small intestinal organoids and promoting the repair of small intestinal defects remains unreported.

The small intestine submucosa (SIS), derived from porcine tissue, is a commercially available biodegradable material with low immunogenicity [26]. SIS has been extensively utilized in various tissue-repair applications owing to its exceptional biocompatibility, bioactivity, absorbability, and capacity to promote autologous tissue growth [27]. In particular, SIS has shown significant utility in gastrointestinal repair, including treatment for ulcerative colitis [28] and colonic anastomotic leakage [29]. In recent years, with the emergence of organoid technology, SIS hydrogel has also been utilized for the cultivation of intestinal organoids [18]. Unlike Matrigel, SIS hydrogel more closely mimics the natural microenvironment of intestinal organoids, which is more conducive to the growth of intestinal organoids [18]. However, challenges in shape control and relatively weaker mechanical properties hinder its potential for clinical translation.

GelMA(Gelatin Methacryloyl), a derivative of gelatin, exhibits an exceptional potential as a biomaterial for tissue regeneration owing to its similarity to the extracellular matrix (ECM) and its adjustable physical and chemical characteristics [30,31]. Numerous investigations have substantiated that GelMA hydrogel represents a promising biomaterial for promoting wound healing [32,33]. In addition to providing a moist and sterile environment for wound healing, the inherent photo-crosslinking property of GelMA hydrogel renders it particularly suitable for irregular wound applications. In recent years, there has been a significant advancement in the utilization of GelMA-based composite hydrogels within the realm of organoids [34,35,36].

Building on this progress, we designed and fabricated FC/SIS/GelMA hydrogels by combining the distinctive strengths of these three materials. The FC/SIS/GelMA hydrogel offers several distinct advantages over Matrigel: (1) it closely replicates the native microenvironment of small intestinal organoids without harboring any uncharacterized risks associated with Matrigel; (2) it features rapid gelation properties; (3) it demonstrates remarkable potential for promoting intestinal defect repair; (4) its mechanical properties can be precisely tailored and controlled; (5) it offers a cost-effective alternative. The feasibility of the in vitro culturing of small intestinal organoids was assessed and compared with the results of Matrigel-based culturing. Subsequently, the FC/SIS/GelMA hydrogel–organoid complex was subcutaneously implanted into nude mice to evaluate its in vivo potential for supporting organoid growth. Finally, we developed an intestinal defect model in Sprague Dawley (SD) rats to investigate the clinical translational potential of the FC/SIS/GelMA hydrogel for small intestine repair. Our study revealed that the FC/SIS/GelMA hydrogel exhibited excellent biocompatibility and effectively supported the proliferation and differentiation of small intestinal organoids both in vivo and in vitro, with no significant disparity compared to organoids cultured in Matrigel. Additionally, the FC/SIS/GelMA hydrogel demonstrated significant efficacy in promoting the repair of small intestinal defects. In conclusion, our study presents a novel approach for the clinical application of small intestinal organoids and establishes an experimental foundation for constructing tissue-engineered small intestines based on small intestinal organoids.

## 2. Results

### 2.1. Construction and Characterization of the FC/SIS/GelMA Hydrogel

The FC/SIS/GelMA hydrogel exhibited an extremely rapid gelation ability under UV irradiation (Figure 1A,E), reducing the gelation time by 65 times when compared to Matrigel. This rapid gelation process allowed for flexible adjustments to its morphology based on the specific requirements of different applications (Figure 1C). Additionally, the morphological integrity of the FC/SIS/GelMA hydrogel remained unaltered over a period of 15 days (Figure 1B).

Then, we investigated the microstructure of the FC/SIS/GelMA hydrogel after gelation. Representative SEM images (Figure 1D) demonstrated that FC/SIS/GelMA hydrogels with different FC concentrations displayed a homogeneous porous structure. As the FC concentration increased, the pore diameters of the hydrogels gradually decreased (Figure 1G). The porosity of the FC/SIS/GelMA hydrogels gradually decreased with increasing FC concentration (Figure 1F), reaching the lowest porosity (50% ± 1%) at an FC concentration of 15 mg/mL. The mechanical properties of the FC/SIS/GelMA hydrogel play a crucial role in governing fundamental cellular processes, including growth, proliferation, migration, differentiation, and organoid formation. Therefore, the assessment of biomaterials should prioritize the evaluation of their mechanical properties as a key determinant. The results demonstrated that the mechanical strength of FC/SIS/GelMA hydrogels exhibited a positive correlation with increasing FC concentration (Figure 2D), thereby indicating the tunable mechanical properties of FC/SIS/GelMA hydrogels. In addition, we assessed the degradation properties of the hydrogel, and observed a decrease in the degradation rate of the FC/SIS/GelMA hydrogels with increasing FC concentration (Figure 2C). The swelling capacity of hydrogel serves as an indicator of its hydrophilicity. As depicted in Figure 2B, it was evident that the augmentation in FC concentration led to a reduction in the rate of hydrogel swelling.

In summary, we found that the hydrogel with a concentration of 5 mg/mL of FC demonstrated an optimal degradation rate, porosity, pore size, mechanical strength (intestinal stem cell expansion is optimal within a matrix of 1.3 kPa stiffness [37], 5 mg/mL FC is closest to this value. It is also closer to the mechanical strength of fresh human normal small intestine (2.6 kPa) [38]), and high hydrophilicity. These characteristics were highly conducive to the formation of crypt-like budding domains in small intestinal organoids as well as nutrient supply. Consequently, we proceeded with subsequent experiments utilizing hydrogel with a concentration of 5 mg/mL of FC.

### 2.2. FC/SIS/GelMA Hydrogel Enabled the Formation of Mouse Intestinal Organoids

Further, we investigated the potential of hydrogel for culturing small intestinal organoids and compared its performance with that of Matrigel. Isolated small intestinal crypts from C57BL/6 mice were dispersed in FC/SIS/GelMA hydrogel as well as Matrigel. As depicted in Figure 3(A1–C4), on the first day of culture, all three groups exhibited spherical structures formed by intestinal crypts. By day three of culture, both the FC/SIS/GelMA hydrogel group and the Matrigel group displayed budding domains resembling crypt-like structures. By day seven, the FC/SIS/GelMA hydrogel group developed small intestinal organoids morphologically similar to those of the Matrigel group. However, their efficiency and diameter size were comparatively lower than those observed in the Matrigel group (Figure 3D,E). Interestingly, we observed that organoids cultured solely with FC/GelMA hydrogel exhibited a spherical morphology without budding structures, and failed to replicate the complex architecture seen in Matrigel (Figure 3(B1–B4)). Moreover, their formation efficiency and diameter were significantly lower compared to the other two groups. Notably, only upon supplementation with SIS did the organoids acquire a morphology resembling that of Matrigel.

### 2.3. Biocompatibility of the FC/SIS/GelMA Hydrogel

Suitable biocompatibility is a prerequisite for the cultivation of small intestinal organoids. The cytotoxicity of FC/GelMA and FC/SIS/GelMA hydrogel was evaluated by a cell Live–Dead staining assay on the fourth day of culture (Figure 4(A1–C3)). The results indicated that all three groups exhibited excellent biocompatibility, with the Matrigel group exhibiting the lowest cytotoxicity and a live cell percentage of 83.6%. The percentage of live cells in the FC/SIS/GelMA hydrogel group was 78.8%, whereas the FC/GelMA group had the lowest percentage of live cells (72.4%), and although there was a slight difference in cell survival among the three groups, the difference in cell survival among the three groups was not statistically significant (Figure 4D). After analyzing the aforementioned findings, we have concluded that the inability of FC/GelMA hydrogel to generate budding structures is not attributed to extensive cell death but, rather, likely results from a deficiency in specific substances or growth factors essential for organoid differentiation.

The experimental results presented in Figure 3 and Figure 4 showed that the FC/GelMA group could not be used to culture small intestinal organoids. As a result, this group was excluded from further research in subsequent experiments.

The immunogenicity of the material is a pivotal determinant influencing the prospective clinical application of hydrogel. Our study demonstrated (Figure 2A) that on the fourteenth day post subcutaneous implantation in C57BL/6 mice, the FC/SIS/GelMA hydrogel group exhibited minimal macrophage infiltration, which was significantly lower than that observed in the Matrigel group (positive control), and slightly higher than that seen in the blank group (negative control). These findings suggest that FC/SIS/GelMA hydrogel induced only a mild inflammatory response and holds great potential for clinical applications.

### 2.4. FC/SIS/GelMA Hydrogel Facilitated the Maintenance of Proliferative Activity in Small Intestinal Organoids

The proliferative activity of organoid cells serves as a crucial determinant for assessing the suitability of a hydrogel in supporting small intestinal organoid culture. To evaluate the maintenance of proliferative capacity in small intestinal organoids cultured within FC/SIS/GelMA hydrogel, we conducted EDU assays on days 1, 5, and 7. As shown in Figure 5B, the cells of small intestinal organoids exhibited robust proliferation and localized primarily in the peripheral bud (crypt) region, aligning with the typical proliferative pattern observed in small intestinal organoids as previously reported [39]. Furthermore, there was no significant disparity in proliferative activity compared to small intestinal organoids cultured in Matrigel (Figure 5A).

### 2.5. The FC/SIS/GelMA Hydrogel Demonstrated the Capacity to Facilitate the Differentiation of Intestinal Crypts into Fully Mature Organoids In Vitro

To further characterize cultured intestinal organoids and the differentiation of intestinal stem cells (ISCs) in FC/SIS/GelMA hydrogel, we selected representative markers specific to intestinal epithelial cells and examined their expression and localization through immunofluorescence analysis (Figure 6A for hydrogel, Figure 6B for Matrigel). In general, organoids embedded in both matrices exhibited comparable morphologies and expressed the targeted cell markers. The presence of epithelial adhesion regions confirmed by E-cadherin expression suggested that intercellular interactions within the organoids were preserved. Intestinal organoids cultured in FC/SIS/GelMA hydrogel exhibited comparable retention of intestinal epithelial cells, including absorptive and secretory cells, to those grown in Matrigel. The positive staining areas for key differentiation marker proteins (MUC2, LYZ, VILLIN) in the FC/SIS/GelMA hydrogel-cultured organoids did not show significant differences compared to the ones cultured in Matrigel.

Quantitative real-time polymerase chain reaction (qPCR) demonstrated that the expression levels of stem cell markers (Lgr5, OLFM4) and goblet cells marker (Muc2) in intestinal organoids encapsulated in FC/SIS/GelMA hydrogel were higher than those embedded in Matrigel (Figure 7). This could be attributed to the superior compatibility of FC/SIS/GelMA hydrogel with the native microenvironment of small intestinal organoids compared to Matrigel. All other intestinal differentiation markers (CHGA for enteroendocrine cells, Lyz1 for Paneth cells, CLDN18 for tight junction protein, and KRT20/Vil1 for enterocytes) of the intestinal organoids grown in FC/SIS/GelMA hydrogel were expressed at comparable levels to those grown in Matrigel.

### 2.6. FC/SIS/GelMA Hydrogel Maintained Organoid Morphology and Function In Vivo

To assess the in vivo performance of FC/SIS/GelMA hydrogel-loaded small intestinal organoids, we first cultured the hydrogel–organoid complex in vitro for three days, followed by subcutaneous transplantation into nude mice. After one week, samples were collected for histological and immunofluorescence staining evaluation.

After one week of in vivo culture, the complexes maintained their fundamental morphology (Figure 8A). The comprehensive H&E staining revealed (Figure 8(C1)) a homogeneous distribution of small intestinal organoids within the hydrogel, exhibiting a remarkable survival rate. The local magnification of H&E staining (Figure 8(C2,C3)) revealed that the small intestinal organoids maintained their cystic architecture with tightly interconnected peripheral cells. Alcian blue staining demonstrated the sustained mucus-secreting function of the small intestinal organoids (Figure 8(B1,B2)). Immunofluorescence staining results (Figure 8D) demonstrated that the small intestinal organoids maintained their differentiated state in vivo, exhibiting the stable presence of various types of intestinal signature cells including goblet cells, Paneth cells, and enterocytes. The detection of E-cadherin indicated the existence of tightly connected cell-to-cell structures, promoting the stability of organoid morphology.

In conclusion, the intestinal organoids exhibited sustained morphological stability, functional integrity, and differentiation characteristics in vivo, thereby establishing a solid experimental foundation for the in vivo application of FC/SIS/GelMA hydrogel.

### 2.7. The FC/SIS/GelMA Hydrogel Exhibited Potential in Facilitating the Regeneration of Intestinal Defects

The schematic diagram of the intestinal repair experiment is depicted in Figure 9A. First, a 1 cm diameter incision was made on the defect of the intestinal wall, which was then sutured with two absorbable stitches. Sprague Dawley (SD) rats were divided into three groups. (1) The control group (no treatment), (2) the SIS/GelMA group (defects covered with SIS/GelMA hydrogel), and (3) the FC/SIS/GelMA hydrogel group (defects covered with FC/SIS/GelMA hydrogel). The SIS/GelMA group served as a control to assess the promoting effect of FC on intestinal defect repair.

Next, we conducted experiments to assess the adhesive properties of FC/SIS/GelMA hydrogel at the site of the intestinal defect. A segment of SD rat intestine was utilized to create the defect which was then covered with hydrogel. The hydrogel’s adhesion was tested through stretching, bending, twisting, and flushing trials. The experimental findings demonstrated the robust attachment of the hydrogel at the intestinal defect site (Figure 9B).

According to the histological staining results of the blank group at 14 days post-operation (Figure 10A), the intestinal repair outcomes were remarkably inadequate, exhibiting minimal regeneration of intestinal villous tissue at the defect site and the formation of substantial and irregularly shaped scar tissue. The wound area exhibited the significant infiltration of inflammatory cells. Masson staining revealed the disorganized distribution of collagen fibers within the scar, while immunohistochemical staining for α-SMA demonstrated an absence of positive α-SMA expression at the site of repair, indicating a lack of smooth muscle tissue regeneration. The SIS/GelMA group exhibited superior reparative efficacy compared to the blank group (Figure 10B), with more regular scar tissue formation and partial coverage of intestinal villus tissue at the repair site. And the migration of epithelial tissue was observed at the wound margins. Masson staining revealed a more uniform distribution of collagen fibers. However, α-SMA regeneration was still not observed. In contrast, the FC/SIS/GelMA hydrogel group exhibited ideal results for intestinal defect repair (Figure 10C), characterized by the significant regeneration of intestinal villous tissue at the defect site, with epithelial tissue covering the entire defect site. Moreover, there were scattered smooth muscle cells with morphologic and staining features consistent with the smooth muscle cells of native small bowel. Also, there were no apparent indications of inflammatory reaction. Masson staining revealed the presence of small scar tissue, while α-SMA immunohistochemical staining demonstrated the uniform production of α-SMA surrounding the defect site, the phenomenon providing evidence for the regeneration of smooth muscle tissue. In conclusion, the FC/SIS/GelMA hydrogel exhibited potential in promoting intestinal defect repair. Compared with the SIS/GelMA group, the FC/SIS/GelMA hydrogel group showed more ideal villus regeneration, smaller scar area, and less cell infiltration, indicating that FC played a role in promoting villus regeneration and inhibiting inflammatory response.

## 3. Discussion

The advancement of intestinal organoid technology offers a novel method for investigating the structural and functional aspects of the intestine, elucidating the mechanisms underlying intestinal diseases, exploring personalized medicine approaches, and advancing intestinal regenerative medicine [40,41]. However, the current sourcing of Matrigel from basement membranes secreted by Engelbreth–Holm–Swarm mouse sarcoma cells, a commonly used material for culturing intestinal organoids, poses limitations on its clinical application due to undetermined components, batch-to-batch variability, and high cost [42].

In recent years, a multitude of scholars have been investigating alternative materials to replace Matrigel for the cultivation of intestinal organoids, with synthetic or natural hydrogels being regarded as an optimal solution. Meghan M. Capeling et al.’s studies have demonstrated that non-adhesive alginate hydrogels are capable of supporting the growth and maturation of human intestinal organoids [17], while Rodrigo Curvello et al.’s [16] research has shown that collagen–nanocellulose hydrogel could provide an affordable, high-performing, thermo-responsive, and sustainable matrix for intestinal organoid growth. Similarly, the study conducted by Ricardo Cruz-Acuna et al. [43] demonstrated that PEG-4MAL hydrogels exhibited robust and highly reproducible in vitro growth and expansion of human intestinal organoids.

However, the aforementioned studies exhibited certain limitations. The success rate of intestinal organoids cultured in alginate was significantly lower than that of Matrigel, and the mortality rate of the organoids at the later stage of culture (after 28 days) was also significantly higher than that of Matrigel. Lower organoid yields in alginate as compared to Matrigel may be due to the inability of cells to remodel the hydrogel, lack of interactions with serum proteins, or the lack of growth factors present in Matrigel. Regarding collagen–nanocellulose hydrogels, dissected crypts directly seeded in collagen–nanocellulose hydrogel do not progress into organoids. Only when a small amount of Matrigel-20% (*v*/*v*) is added to the hydrogel will fresh intestinal crypts form organoids. It could also be a consequence of the lack of certain growth factors in the hydrogels. As for PEG-4MAL hydrogels, an important aspect of this engineered hydrogel platform is its rapid reaction kinetics, which, if mixing is not conducted properly, can lead to the formation of an inhomogeneous gel that presents variability in its physicochemical properties. This seriously limits the feasibility of its clinical application.

To address these limitations, we developed and synthesized a novel hydrogel that closely mimicked the native microenvironment of small intestinal organoids. This hydrogel not only facilitated the proliferation and differentiation of small intestinal organoids but also enhanced the regenerative capacity for repairing intestinal defects, thereby offering a promising approach for clinical applications of small intestinal organoids.

The application prospects of fish collagen are extensive, encompassing various clinical disciplines such as burns, trauma, corneal diseases, cosmetology, orthopedics, artificial blood vessel construction, and wound hemostasis [44,45]. In the area of wound healing [22], a chitosan/fish collagen wound dressing received approval in China in 2013. In the same year, a “cod skin acellular matrix” wound dressing composed mainly of collagen was approved by the US FDA. Additionally, Eucare Pharmaceuticals (India) has received CE approval for marketing their fish collagen hemostatic product which can be used in various wound-care repairs. However, there have been no reports on the potential use of fish collagen for intestinal defect repair and small intestinal organoid culture.

Its excellent biocompatibility, bioactivity, absorbability, low immunogenicity, and strong ability to promote autologous tissue growth have rendered SIS a subject of significant interest in the field of tissue repair [46]. Since the introduction of the first SIS-derived product approved by the FDA, SIS grafts have gained widespread utilization in tissue regeneration and repair across diverse clinical applications [26,47,48]. In recent years, with the emergence of organoid technology, SIS has also found application in the culture of organoids. A study conducted by Suran Kim et al. [49] demonstrated that intestinal organoids cultured using SIS hydrogel exhibited comparable growth, development, and functionality to those cultured with Matrigel. Consequently, SIS holds potential for culturing small intestinal organoids. However, its limited mechanical properties and prolonged gelation time hinder its clinical applicability.

GelMA hydrogels exhibit a broad spectrum of applications in the field of tissue engineering, encompassing the regeneration or repair of bone, cartilage, nerves, skin, skeletal muscle, cardiac tissue, and the liver [31,50,51]. They are particularly advantageous for wound healing due to their resemblance to the extracellular matrix (ECM) and their customizable physical and chemical properties. Recently, GelMA-based hydrogels have also been employed for the cultivation of diverse organoids, including intrahepatic cholangiocyte organoids [35], adipose tissue organoids [52], and islet organoids [34].

In this study, we developed FC/SIS/GelMA hydrogels by synergistically integrating the unique merits of these three materials. The resultant hydrogel possessed several distinct advantages over Matrigel: (1) it closely mimicked the native microenvironment of small intestinal organoids without harboring any uncharacterized risks associated with Matrigel; (2) it exhibited rapid gelation properties; (3) it demonstrated remarkable potential for promoting intestinal defect repair; (4) its mechanical properties could be precisely tailored and controlled; (5) it offered a cost-effective alternative. It is noteworthy that our study revealed the limited formation of budding structures in small intestinal organoids cultured with FC/GelMA hydrogel, which could only be achieved when SIS was added to promote the ideal formation of small intestinal organoids. The analysis suggested that this limitation might arise from the absence of essential components and growth factors necessary for organoid development. The phenomenon observed in this study was consistent with the findings of Joo Hyun Jee et al. [53], suggesting that application type I collagen hydrogel-cultured intestinal organoids exhibited limited budding structures, while expressing genes, proteins, and cells associated with the development of small intestinal organoids. Similarly, Ziyad Jabaji MD et al. [54] demonstrated that type I collagen-cultured intestinal organoids formed smooth luminal structures resembling the intestine. However, co-culture with ISEMF in vivo restored budding structures. The primary component of the fish collagen utilized in this study was type I collagen, thus suggesting that type I collagen might also serve as a contributing factor to this phenomenon.

Due to the unique photo-crosslinking characteristics of GelMA, the FC/SIS/GelMA hydrogel exhibited rapid gelation ability, leading to a significant reduction in operation time (by 65 times) and operation complexity compared to Matrigel. This feature enhanced its potential for widespread adoption and application. The FC/SIS/GelMA hydrogels possessed a porous structure, which conferred advantages in facilitating the acquisition of sufficient oxygen and nutrients by the implanted organoids. Furthermore, the mechanical properties of FC/SIS/GelMA hydrogels exhibited a positive correlation with the concentration of fish collagen, thereby demonstrating their inherent tunability and potential suitability for diverse clinical applications. The hydrogels exhibited favorable hydrophilic and degradation properties, thereby providing a physical assurance for the growth, development, and clinical translation of small intestinal organoids.

The clinical translation of hydrogel necessitates impeccable biocompatibility, which serves as a fundamental prerequisite [55]. Although each individual material had independently demonstrated favorable biocompatibility, it remained imperative to experimentally validate whether the amalgamation of these three materials upheld such desirable biocompatibility [56,57,58]. In this study, the FC/SIS/GelMA hydrogel exhibited low cytotoxicity and immunogenicity. The intestinal organoids embedded in the FC/SIS/GelMA hydrogel maintained high cellular activity and proliferation ability, while exhibiting a growth process and morphological appearance similar to that of the Matrigel. Therefore, FC/SIS/GelMA hydrogel demonstrated its potential for the cultivation of small intestinal organoids and clinical applications.

The ability to culture mature small intestinal organoids is a crucial criterion for assessing the potential application of hydrogels. Our study demonstrated that the FC/SIS/GelMA hydrogel facilitated the successful differentiation of intestinal crypts into fully mature intestinal organoids, characterized by the expression of specific cell markers including enterocytes, Paneth cells, and goblet cells. The qPCR results further confirmed these findings, demonstrating the expression of various cell signature genes. Notably, in FC/SIS/GelMA hydrogel, the expression levels of LGR5, OLFM4, and MUC2 were significantly higher compared to Matrigel. This observation suggested that the enhanced composition of the FC/SIS/GelMA hydrogel better mimicked the native microenvironment of small intestinal organs, rendering it more conducive to the growth and differentiation of small intestinal crypt stem cells. Consistent with our findings, a subsequent study also demonstrated the viability of utilizing an intestinal acellular matrix for culturing small intestinal organoids. In line with our findings, a study by Giovanni Giuseppe Giobbe et al. [18] demonstrated that intestinal extracellular matrix (ECM) gels possessed the capability to support the culture of not only intestinal organoids but also cells derived from other endoderm-derived tissues, such as the liver, stomach, and pancreas. Additionally, Zi-Yan Xu et al. [59] fabricated a dECM-GelMA bioink with tunable mechanical properties, which effectively facilitated the proliferation of intestinal stem cells and the formation of organoids. Notably, we found a lower efficiency and diameter of organoids cultured from FC/SIS/GelMA than Matrigel. We posited that the enhanced mechanical properties exhibited by the FC/SIS/GelMA hydrogel imposed constraints on the growth and proliferation of organoids.

The in vivo performance of hydrogels in animals serves as a crucial indicator of their clinical translatability. To assess the viability and differentiation of hydrogel–organoid complexes in vivo, we conducted transplantation experiments in subcutaneous pockets of immunodeficient mice. The subcutaneous culture of hydrogel–organoid complexes in nude mice for one week demonstrated the robust maintenance of small intestinal organoids’ activity and morphology, preservation of mucus secretion function, as well as the consistent presence of various marker cells. The experimental findings demonstrated that the FC/SIS/GelMA hydrogel possessed the capability to sustain the activity and functionality of small intestine organs in vivo, thereby exhibiting promising potential for clinical translation.

The repair of intestinal defects poses a significant challenge that needs to be addressed. In order to tackle this issue, we have developed an innovative animal model for studying intestinal defects and exploring the potential clinical application of FC/SIS/GelMA hydrogels in facilitating intestinal repair. Our research demonstrated that the FC/SIS/GelMA hydrogel exhibited potential in facilitating intestinal defect repair, accelerating the regenerative process, mitigating scar tissue formation, and promoting intestinal epithelial migration and regeneration, as well as enhancing small intestinal villus and smooth muscle regeneration. Moreover, the hydrogels additionally exhibited anti-inflammatory properties. The combined action of fish collagen and SIS might contribute to this phenomenon, as evidenced by numerous reports highlighting the beneficial impact of fish collagen on wound repair across various contexts. Numerous studies have demonstrated that fish collagen and its hydrolysate possess the ability to effectively facilitate intestinal defect repair and re-epithelialization processes [60]. The study conducted by Zhao Deng et al. [61] demonstrated that FSGHF3 and peptides (isolated from fish skin gelatin hydrolysate) could alleviate intestinal inflammation and maintain the intestinal barrier. Mouna Rahabi [62] identified fish collagen peptides as a protective agent against colitis directly acting on macrophages by orienting their polarization toward an anti-inflammatory, immunotolerant, and antioxidant phenotype in an MR-dependent manner. Additionally, he demonstrated that, through their effect on the immune system, fish collagen peptides maintained intestinal eubiosis. The study conducted by K. Sivaraman and C. Shanthi [25] demonstrated that the purified hydrolysate fraction of fish skin collagen exhibited potential as a nutraceutical agent for the prevention or treatment of internal intestinal ulcer wounds. Similarly, SIS has demonstrated its potential in promoting the repair of gastrointestinal defects. Keane et al. [28] demonstrated that SIS hydrogels exhibited therapeutic potential for the treatment of ulcerative colitis, facilitating enhancement in colonic epithelial barrier function and the resolution of the proinflammatory state of tissue macrophages. Henry Huson, MD et al. [29] employed a Polyvinyl Alcohol-Porcine Small Intestine Submucosa Stent for the treatment of anastomotic leakage in the porcine colon, and the study demonstrated the absence of any post-operative complications.

Currently, some limitations of our trial should be considered. First, further optimization is required for the material’s optimal concentration. While the hydrogel mimicked the structure of the ECM to some extent, its mechanical properties were greater than those of the intestine. Excessive mechanical strength hindered the growth of small intestinal organoids, whereas insufficient mechanical strength compromised intestinal repair. Therefore, future experiments should focus on adjusting the concentration ratios of the three components to determine the optimal concentration of the hydrogel formulation. Second, additional experimental verification is needed to determine if FC/SIS/GelMA hydrogel-loaded intestinal organoids can better promote intestinal defect healing. Third, our study is limited to rodents; therefore, further investigations are necessary to evaluate the long-term performance of FC/SIS/GelMA hydrogel in larger animals. Fourth, further investigation should be conducted to identify the specific component that facilitated the formation of budding structures. Lastly, compared with Matrigel, the diameters of the cultured organoids in FC/SIS/GelMA hydrogel were relatively small, and, analyzing the reasons, we believe that the main reason is that the mechanical properties of FC/SIS/GelMA hydrogel are higher than that of Matrigel, and a larger mechanical strength is not conducive to the growth of the organoids, which restricts the diameter of the organoids. But a too-low mechanical strength is not conducive to the control of the morphology of the hydrogel as well as to the promotion of the gut defect repair; therefore, in subsequent experiments, optimizing the concentration ratio of the three components and exploring the most ideal hydrogel concentration ratio remains essential.

Although this experiment still possesses the aforementioned limitations, FC/SIS/GelMA hydrogel also exhibits significant potential for various applications. For instance, when combined with 3D printing technology, shape-adjustable hydrogel can be engineered to exhibit enhanced suitability for clinical applications [63]. Combined with tissue-engineering technology, the utilization of small intestinal organs and hydrogel presents a promising approach for the in vitro regeneration of small intestine tissue [64]. For example, Satoshi Watanabe et al. infused colonic organoid tissue into the luminal space via the anus. The infused organoids subsequently attached to the injured region and rebuilt a donor-derived epithelium. This study provides an experimental basis for the treatment of refractory ulcerative colitis using colonoid organ transplantation therapy [11]. If FC/SIS/GelMA hydrogel can be co-cultured with colonic organoids in vitro and then injected into the damaged intestinal tract, in situ crosslinking will enable it to be firmly affixed to the surface of the damaged intestinal tract. This will bring the following benefits: firstly, the FC/SIS/GelMA hydrogel has the effect of promoting the repair of intestinal defects, and secondly, it can make the organoids tightly bond to the damaged site, providing an ideal carrier for the organoids. This will significantly improve the therapeutic effect of colitis.

In summary, this study presents a novel approach for the cultivation and clinical application of small intestinal organoids, thereby offering a promising pathway for the advancement of tissue-engineered intestine.

## 4. Materials and Methods

### 4.1. Materials

Fish collagen was purchased from Macklin (Shanghai, China). Gelatin methacryloyl (60% degree of substitution) was purchased from Engineering For Life (Suzhou, China). Matrigel was purchased from Mogengel (Xiamen, China). Porcine small intestinal submucosa powder (SIS) was purchased from Sun Shing (Guangzhou, China)

### 4.2. Gelation Protocol

For gelation, we followed the protocols established in previous studies [18], the SIS powder was digested in pepsin/HCl solution (30 mg/mL SIS in 1 mg/mL 0.1 M HCl) at RT for 72 h with constant rotation until complete dissolution. Once dissolved, the solution was neutralized to a physiological pH of 7.4 by adding NaOH (10 M). The resulting pregel (SIS concentration: 30 mg/mL) was then stored at 4 °C to prevent unexpected gelation. A similar process was employed for the preparation of FC solutions, but with concentrations of 15 mg/mL, 30 mg/mL, and 45 mg/mL.

### 4.3. Synthesis of the FC/SIS/GelMA Hydrogel

The results of previous studies have demonstrated that a GelMA concentration of 5% is the most suitable for cell culture and growth, while an SIS concentration of 10 mg/mL is optimal for supporting the growth of small intestinal organoids [59]. Building on these findings, our study aimed to identify the optimal concentration of fish collagen. The 3 × 0.25% (*w*/*v*) lithium phenyl-2,4,6-trimethylbenzoylphosphinate (LAP) standard solution was prepared by dissolving 150 mg of LAP in 20 mL of phosphate-buffered saline (PBS) at a temperature of 40–50 °C for a duration of 15 min. The solution (1 mL) was supplemented with 150 mg of GelMA and stirred at a temperature range of 40–50 °C for 20–30 min until complete dissolution. The GelMA solution (150 mg/mL), SIS pregel (30 mg/mL), and FC solution (15 mg/mL, 30 mg/mL, 45 mg/mL) were subsequently combined in a volumetric ratio of 1:1:1 and vigorously stirred using magnetic stirring to obtain FC/SIS/GelMA mixtures with FC concentrations of 5 mg/mL, 10 mg/mL, and 15 mg/mL. The solution was sterilized using a 0.22 μm sterile injection filter and stored while protected from light. The pre-cured gel solution was then injected into a customized 3D cylindrical mold (diameter: 2 cm, thickness: 5 mm) and exposed to UV light for 20 s to obtain the FC/SIS/GelMA hydrogel. The final concentrations of the hydrogels are presented in Table 1.

### 4.4. Mechanical Properties

The mechanical properties of the hydrogel were assessed using an electronic universal testing machine (Beijing Zhong Ke Bai Ce, Beijing, China). Cylindrical hydrogel samples with dimensions of 20 mm in diameter and 5 mm in height were positioned at the center of the horizontal measuring platform. The compression testing was conducted using a WDT-0.2 mechanical testing instrument, applying a strain rate of 2 mm/min, and the data were continuously recorded until a deformation rate of 50% was achieved. All experiments were performed in triplicate (n = 3).

### 4.5. SEM Analysis

Hydrogel samples were prepared (2 cm in diameter and 5 mm in height). The samples were frozen at −80 °C and then freeze-dried for 48 h. The specimens were cracked, sputter-coated with Pt, and examined using SEM (S-3400N; Hitachi, Tokyo, Japan).

### 4.6. Swelling Property [65]

The swelling property of a hydrogel was assessed using the weighing method. The prepared samples were freeze-dried, and their dry weights were recorded using an electronic balance. Afterward, the samples were immersed in PBS solvent for 24 h and re-weighed. This allowed for analysis of the swelling behavior and calculation of the swelling ratio. The swelling ratio can be determined using the following formula:$$swelling\;ratio=\frac{wet\;weight-dry\;weight}{dry\;weight}\;\times 100\%$$

### 4.7. Degradation [66]

The enzymatic degradation properties of the different hydrogels were assessed by incubating hydrogel samples in DPBS containing 2.5 U mL^−1^ of collagenase type II at 37 °C. At predetermined time intervals, the samples were retrieved from the collagenase type II solution, rinsed with deionized water, frozen at −80 °C, and subsequently lyophilized. The degradation percentage (D%) was determined by establishing a relationship between the mass of the samples at each time point (Wt) and the initial dry weight (W0), as described by the following equation:$$D(\%)=\frac{W0-Wt}{W0}\times 100\%$$

### 4.8. Porosity [67]

The porosity of each hydrogel was assessed using the ethanol infiltration method. V1 and V2 represent the initial and final volumes of ethanol in the measuring bottle, respectively, before and after immersing the scaffolds in ethanol. The remaining volume (V3) was determined by subtracting the volume displaced by the scaffold after 15 min of immersion. The porosity of the scaffolds was calculated using the following equation:$$Porosity(\%)=\frac{V1-V3}{V2-V3}\times 100\%$$

### 4.9. Isolation of Small Intestinal Crypts

All animal protocols were approved by the Animal Ethics Committee of Chongqing Medical University, and conducted at the Animal Experimental Centre of the same institution, in accordance with the National Institute of Health Guide for the Care and Use of Laboratory Animals. Female C57BL/6j mice (4 weeks old) were euthanized and a 5–6 cm segment of the small intestine was excised near the stomach in a sterile environment. The mesentery and adipose tissue surrounding the intestine was meticulously dissected using forceps. After gently scraping the intestinal villi using a surgical blade, the intestinal lumen was thoroughly washed with DPBS at 4 °C. The cleaned small intestine was then sectioned into approximately 2 mm segments, which were further subjected to repeated gentle washing (15–20 times) with DPBS supplemented with antibiotics. Next, the intestinal segments were subjected to digestion by the addition of 10 mL of DPBS containing 5 mM EDTA and incubated at a temperature of 4 °C for approximately 30 min. After digestion, the supernatant containing EDTA was discarded. The crypts were then separated by repeated blowing and resuspension of the tissue fragments in DPBS containing 0.1% BSA, followed by filtration and collection using a 70 μm filter for 3–4 cycles. The resulting tissue suspension was centrifuged to discard the supernatant, yielding a final suspension enriched with small intestinal crypts.

### 4.10. In Vitro and In Vivo Culture of Organoids

The small intestinal crypts were incorporated into the FC/SIS/GelMA hydrogel and Matrigel, respectively. A total of 500 crypts/30 μL of the FC/SIS/GelMA hydrogel or Matrigel mixture were seeded in a 24-well plate. After gelation, the crypts were incubated in IntestiCult™ Organoid Growth Medium (Mouse, STEMCELL Technologies, Vancouver, BC, Canada) and maintained at 37 °C with 5% CO_2_. A portion of the organoids were cultured in vitro for 7 days. The remaining organoids were cultured in vitro for 3 days before being subcutaneously implanted into Balb/c nude mice (lacking the thymus gland) for an additional 7 days. Following these procedures, comprehensive histological analysis was conducted on all samples.

### 4.11. Organoid Viability and Proliferation in Scaffolds

The viability of the small intestinal organoids was assessed at 1, 5, and 7 days using the Live–Dead Cell Viability Assay kit (Beyotime, Shanghai, China). The proliferation of the small intestinal organoids was evaluated at 1, 5, and 7 days using the BeyoClick™ EdU Cell Proliferation Kit with Alexa Fluor 488 (Beyotime, Shanghai, China).

### 4.12. RNA Extraction and qPCR Analysis

Total RNA was extracted using Trizol reagent (Takara Biomedical Technology, Kusatsu, Japan), followed by reverse transcription into single-stranded cDNA and subsequent amplification with specific primers. For qRT-PCR amplification, a 10 μL real-time polymerase chain reaction system (Takara Biomedical Technology, Japan) was employed with 5 μL of SYBR Green PCR Master Mix (Takara Biomedical Technology, Japan), 2 μL of nuclease-free H_2_O, 2 μL of primer, and 1 μL of cDNA. The relative mRNA expression level was normalized to GAPDH. Table 2 provides an overview of the primer sequences.

### 4.13. Whole-Mount Immunofluorescent Staining

The organoids or frozen sections were fixed with 4% paraformaldehyde at 20 °C for 15 min, followed by PBS washing (10 min × 3) and permeabilization with 0.5% (*v*/*v*) Triton X-100 (Beyotime, Shanghai, China) at room temperature. The organoids/frozen sections were then blocked with Immunol Staining Blocking Buffer (Beyotime, Shanghai, China) for 60 min and incubated overnight at 4 °C with primary antibodies. The following primary antibodies were employed (overnight incubation in 4 °C): rabbit MUC2 polyclonal antibody (Proteintech, Wuhan Sanying, Wuhan, Hubei, China), rabbit anti-villin1 antibody (HUABIO, Hangzhou, Zhejiang, China), rabbit anti-E-cadherin antibody (Proteintech, Wuhan Sanying, Wuhan, Hubei, China), rabbit anti-lysozyme antibody (Proteintech, Wuhan Sanying, Wuhan, Hubei, China). The cells/tissues were then stained with fluorescent secondary antibodies such as Goat Anti-Rabbit IgG (AlexaFluor488, Beyotime, Shanghai, China) for 1 h. The nuclei were counterstained with DAPI (Beyotime, Shanghai, China) for 15 min. Following standard immunostaining protocols, fluorescence microscopy was employed to acquire images of the organoids/frozen sections, which were subsequently observed using a Leica microscope (Wetzlar, Germany).

### 4.14. In Vivo Immunological Assessment

Four-week-old male C57BL/6 mice (n = 12) were anesthetized using 1% pentobarbital sodium (50 mg/kg) injection. The mice were subjected to a 0.5 cm incision on their dorsal region. A 200 μL FC/SIS/GelMA hydrogel was crosslinked into a cylindrical shape using UV light irradiation and subsequently implanted beneath the dorsal skin. Similarly, a 200 μL Matrigel was subcutaneously implanted beneath the dorsal skin, followed by the closure of the incisions using sutures. The control group solely received suture treatment. On Day 14, all mice were euthanized by cervical dislocation and the peri-graft tissue was fixed in a 4% paraformaldehyde solution for subsequent immunostaining analysis. (F4/80 Polyclonal antibody, Proteintech, Sanying, Wuhan, Hubei, China).

### 4.15. Repair of Small Intestinal Defects

The nine-week-old male Sprague Dawley rats (n = 30) were anesthetized with a 1% solution of sodium pentobarbital (100 mg/kg). The abdominal area of the rats was shaved and sterilized, followed by the opening of the abdominal cavity to expose the small intestine. The wall of the small intestine was incised with a 1 cm diameter wound, followed by the suturing of two stitches. The blank group received no treatment, while one group was covered with SIS/GelMA hydrogel (10 mg/mL SIS, 50 mg/mL GelMA) and the other group was covered with FC/SIS/GelMA hydrogel. The hydrogels were drip-added to the wound site and then crosslinked in situ. The animals were euthanized for evaluation at a post-operative period of 2 weeks.

### 4.16. Histological and Immunohistochemical Analysis

The samples were fixed in 4% paraformaldehyde, embedded in paraffin and sectioned. Each section was subjected to hematoxylin and eosin (H&E) as well as Masson’s staining for the assessment of small intestine wound healing. Immunohistochemical staining was performed to determine the expression level of α-SMA.

### 4.17. Statistical Analysis

Statistical analyses and graphical representations of the data from this study were mostly generated using GraphPad Prism 8 (GraphPad Software, La Jolla, CA, USA). Results are presented as the mean ± S.D. Unpaired, two-sided Student’s *t*-tests with 95% and 99% confidence intervals were used to determine the significance of the data between the two groups. One-way analysis of variance was conducted to determine the significance of data with more than two groups and was followed by Tukey’s multiple comparisons test. Throughout the study, the sample size was determined based on our preliminary studies and on the criteria in the field. At least three biological samples were included for one experiment and one to three independent experiments were performed to ensure sufficient reproducibility of the results.

## 5. Conclusions

In this study, we developed a photo-responsive hydrogel based on FC/SIS/GelMA featuring adjustable mechanical properties and excellent biocompatibility, which effectively facilitated the proliferation of intestinal stem cells and formation of intestinal organoids. The intestinal organoids embedded within the hydrogel demonstrated enhanced stem cell characteristics, structural integrity, and optimal levels of differentiation while preserving their structural and functional stability in vivo. Furthermore, the FC/SIS/GelMA hydrogel exhibited a significant enhancement in the repair process of intestinal defects. In conclusion, our study provided compelling evidence that FC/SIS/GelMA hydrogel not only facilitated the proliferation and differentiation of small intestinal organoids but also effectively accelerated the regeneration of intestinal defects. This novel approach offers an alternative to Matrigel for culturing organoids and presents a promising avenue for clinical applications. Moreover, our experimental findings establish a solid foundation for the future advancement of organoid-based tissue engineering in small bowel regeneration.

## Figures and Tables

**Figure 1 ijms-26-00663-f001:**
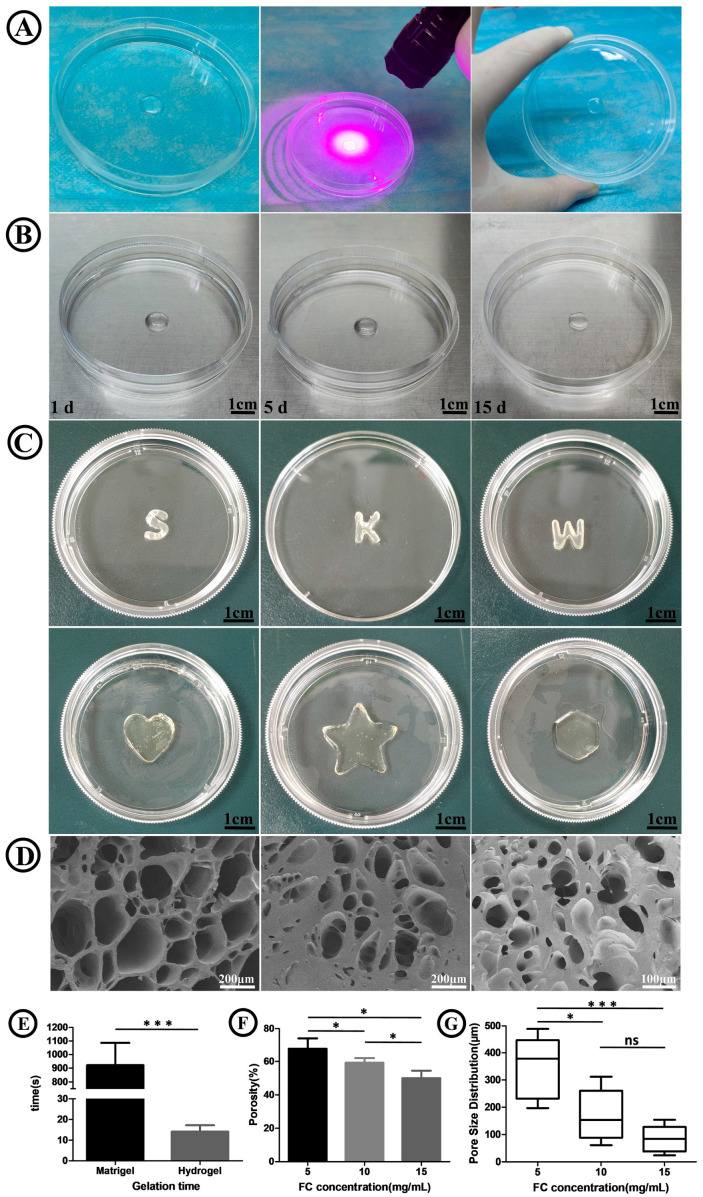
Characterization of FC/SIS/GelMA hydrogel. (**A**) Gelation of FC/SIS/GelMA hydrogels under UV illumination. (**B**) FC/SIS/GelMA hydrogel stability. (**C**) Tunable morphology of FC/SIS/GelMA hydrogels. (**D**) Scanning electron microscope images of hydrogel. (**E**) Comparison of gelation time between FC/SIS/GelMA hydrogel and Matrigel (*n* = 5, *** *p* < 0.001). (**F**) Porosity (n = 5, * *p* < 0.05). (**G**) Internal pore-size distribution (n = 5, * *p* < 0.05, *** *p* < 0.001, ns, no significance.).

**Figure 2 ijms-26-00663-f002:**
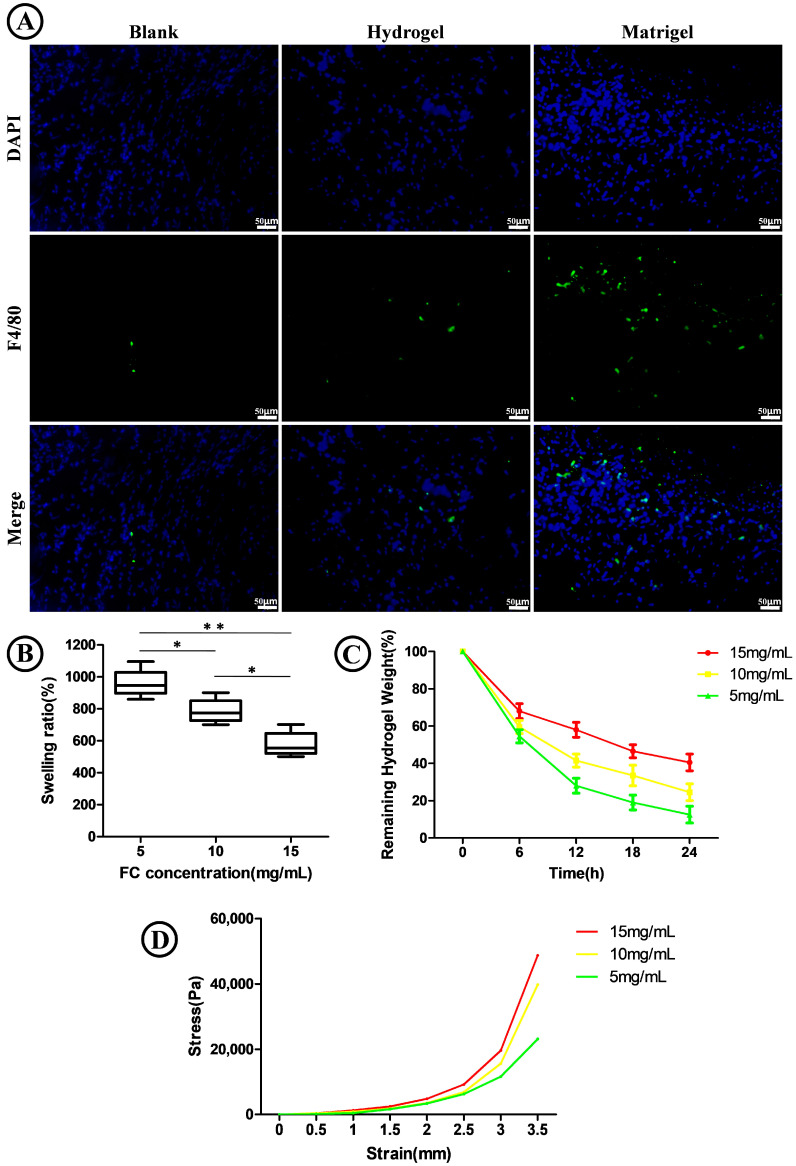
Properties of FC/SIS/GelMA hydrogels. (**A**) Immunofluorescence of tissue around grafts at Day 14. Showing macrophage marker F4/80. Scale bar 50 μm. (**B**) Swelling behavior (n = 5, * *p* < 0.05, ** *p* < 0.01). (**C**) Degradation of FC/SIS/GelMA hydrogels at different concentrations (**D**) Compression test curve.

**Figure 3 ijms-26-00663-f003:**
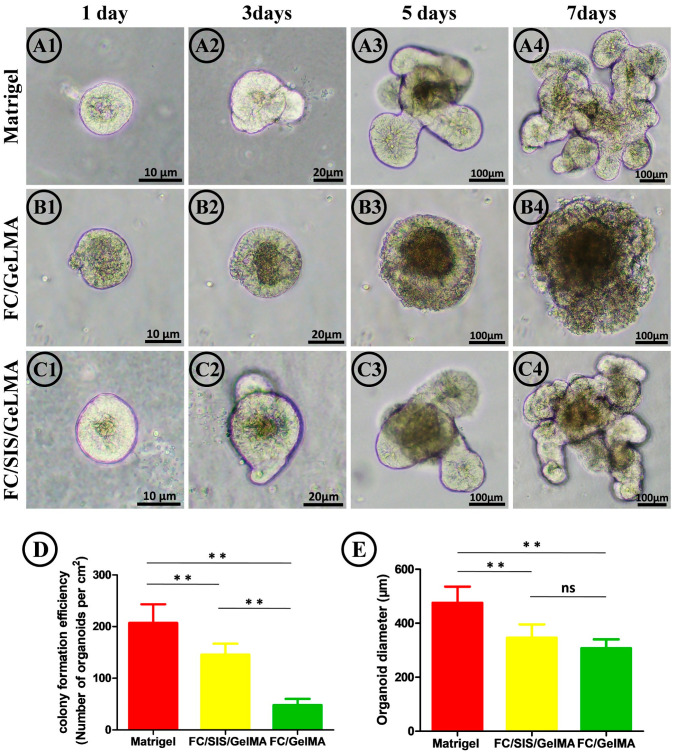
Mouse small intestinal organoids were cultured in FC/SIS/GelMA hydrogel, FC/GelMA hydrogel, and Matrigel. (**A1**–**A4**) Typical bright field images of formed organoids at Days 1, 3, 5, and 7 within Matrigel. (**B1**–**B4**) Typical bright field images of formed organoids at Days 1, 3, 5, and 7 within FC/GelMA hydrogel. (**C1**–**C4**) Typical bright field images of formed organoids at Days 1, 3, 5, and 7 within FC/SIS/GelMA hydrogel. (**D**) Number of intestinal organoids formed on each square centimeter by Day 7 (n = 5, ** *p* < 0.01). (**E**) Analysis of organoids diameters on Day 7 (n = 5, ** *p* < 0.01, ns, no significance.).

**Figure 4 ijms-26-00663-f004:**
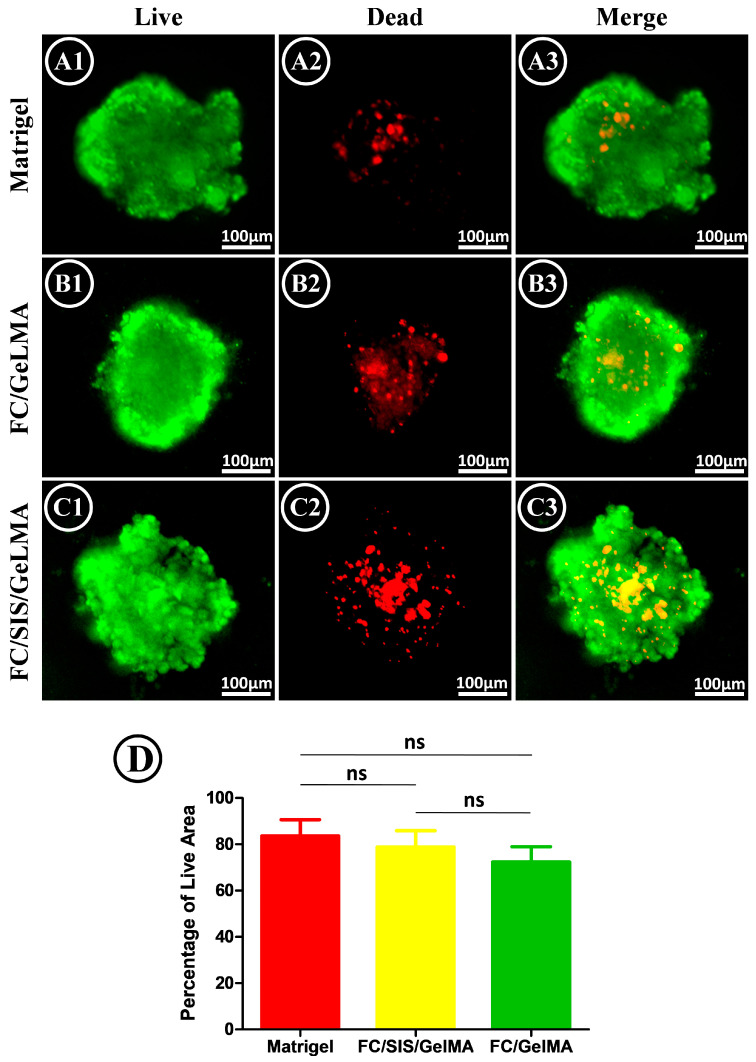
Organoid activity in different hydrogels. (**A1**–**C3**) Live–Dead cell staining to detect the number of live/dead organoids cultured on different hydrogels. (Scale bar = 100 μm). (**D**) Proportion of living cells (n = 5, ns, no significance.).

**Figure 5 ijms-26-00663-f005:**
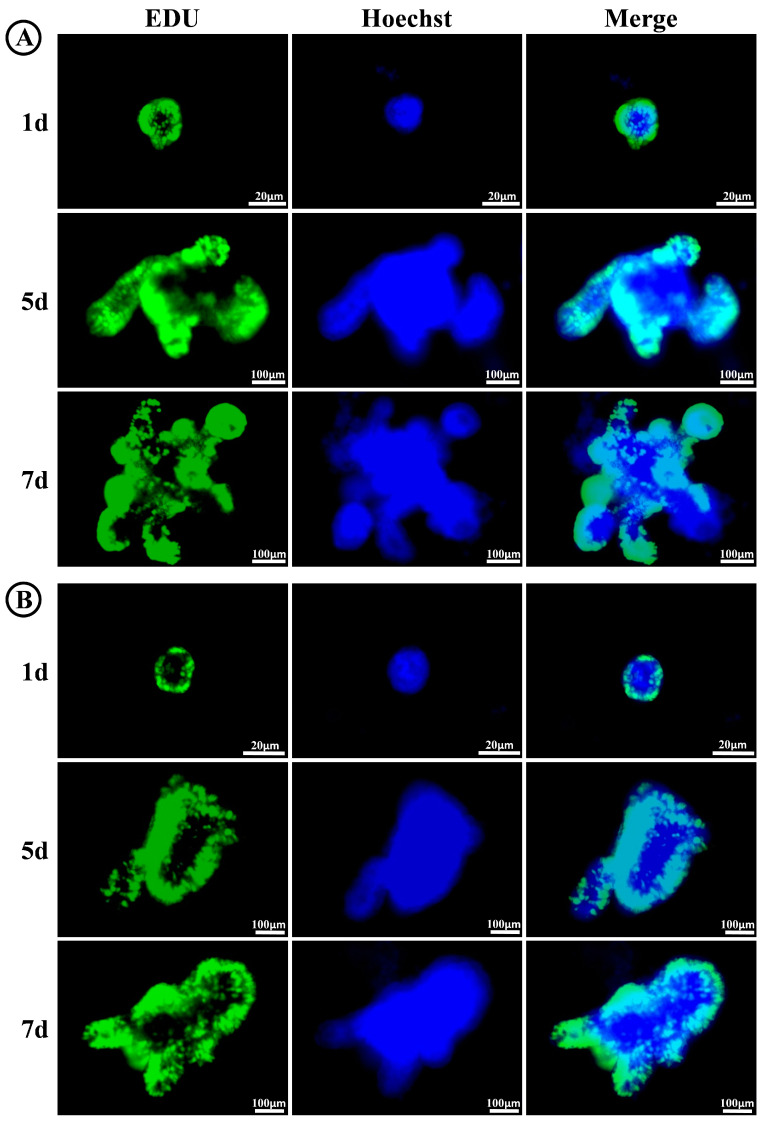
Proliferation of small intestinal organoids in Matrigel and hydrogel. (**A**) EDU staining of small intestinal organoids in Matrigel. (**B**) EDU staining of small intestinal organoids in FC/SIS/GelMA hydrogel.

**Figure 6 ijms-26-00663-f006:**
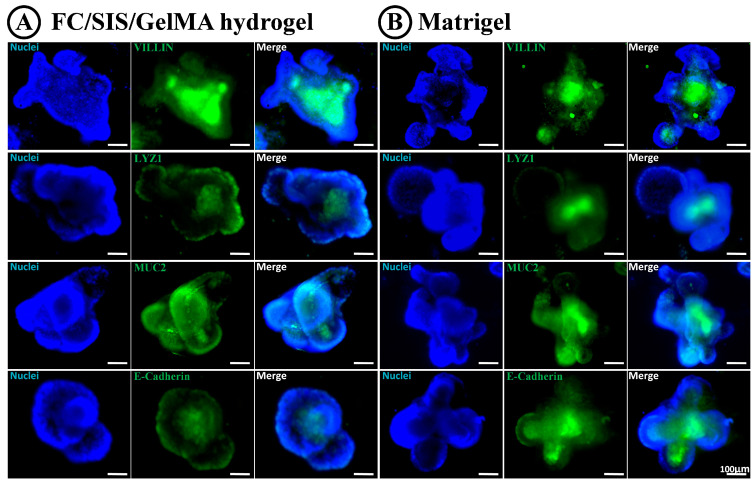
Differentiation pattern of intestinal organoids in FC/SIS/GelMA hydrogel and Matrigel. (**A**) Whole-mount immunofluorescence of cultured organoids on Day 7 in FC/SIS/GelMA hydrogel. (**B**) Whole-mount immunofluorescence of cultured organoids on Day 7 in Matrigel. Scale bar = 100 μm; enterocyte markers = VILLIN; Paneth cell marker lysozyme = LYZ; goblet cell marker mucin-2 = MUC-2; cell–cell adhesion/interaction marker = ECAD.

**Figure 7 ijms-26-00663-f007:**
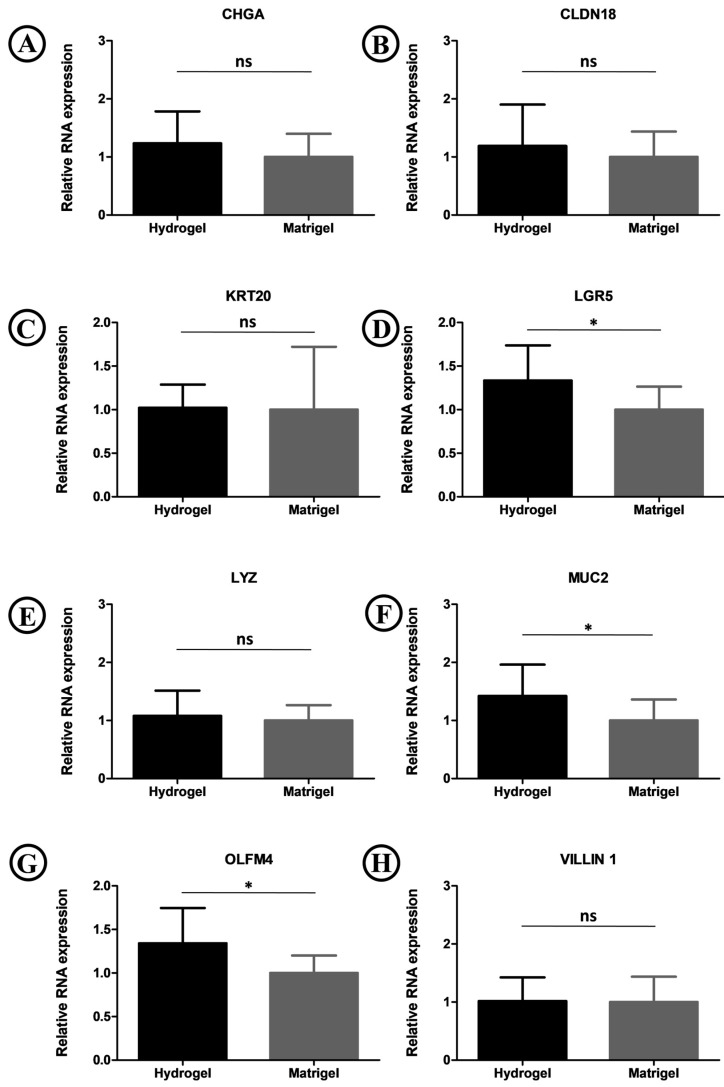
qPCR analysis was conducted to compare the mRNA expression of intestinal organoids embedded in Matrigel and FC/SIS/GelMA hydrogel. * *p* < 0.05, ns, no significance, n = 3. (**A**) Comparison of the expression of CHGA in FC/SIS/GelMA hydrogel and matrige; (**B**) Comparison of the expression of CLDN18 between FC/SIS/GelMA hydrogel and matrigel; (**C**) Comparison of the expression of KRT20 between FC/SIS/GelMA hydrogel and matrigel; (**D**) Comparison of the expression of LGR5 between FC/SIS/GelMA hydrogel and matrigel; (**E**) Comparison of the expression of LYZ between FC/SIS/GelMA hydrogel and matrigel; (**F**) Comparison of the expression of MUC2 between FC/SIS/GelMA hydrogel and matrigel; (**G**) Comparison of the expression of OLFM4 between FC/SIS/GelMA hydrogel and matrigel; (**H**) Comparison of the expression of VILLIN1 between FC/SIS/GelMA hydrogel and Matrigel.

**Figure 8 ijms-26-00663-f008:**
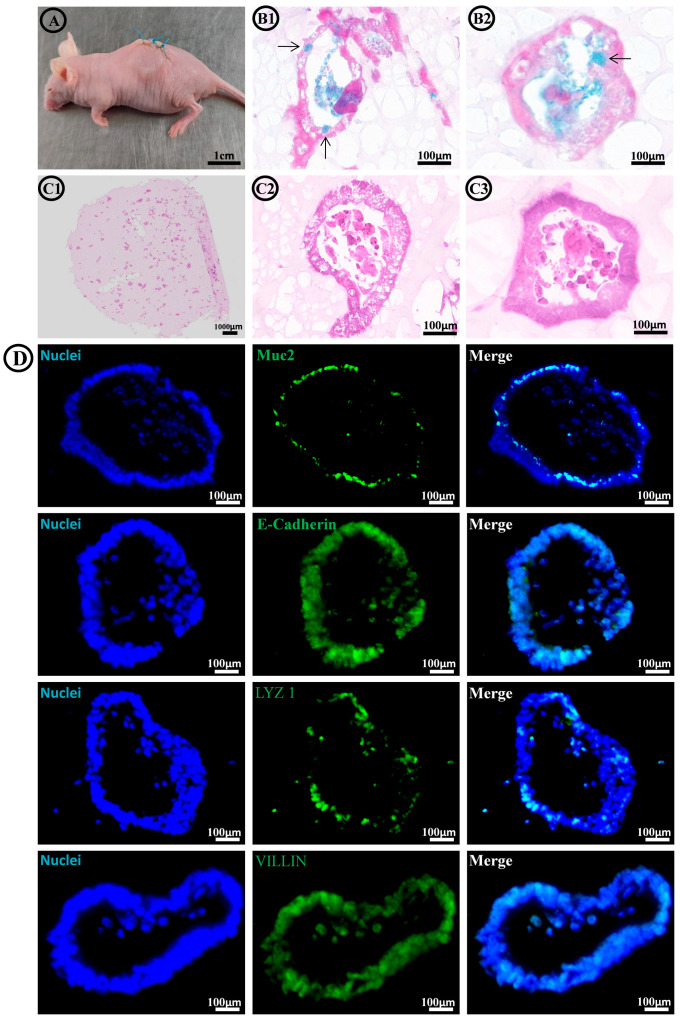
FC/SIS/GelMA hydrogel and Matrigel containing organoids were transplanted subcutaneously into NSG mice. (**A**) Photographs showed external appearance after one week of in vivo incubation. (**B1**,**B2**) Alcian blue staining of small intestinal organoids. Black arrow refers to goblet cells. Scale bars = 100 μm. (**C1**,**C2**,**C3**) Representative microscopic images of H&E staining of small intestinal organoids. Scale bars = 100 μm. (**D**) Immunofluorescence of small intestinal organoids after 7 days of in vivo culture. Scale bar = 100 μm; enterocyte markers = VILLIN; Paneth cell marker lysozyme = LYZ; goblet cell marker mucin-2 = MUC-2; cell–cell adhesion/interaction marker = ECAD.

**Figure 9 ijms-26-00663-f009:**
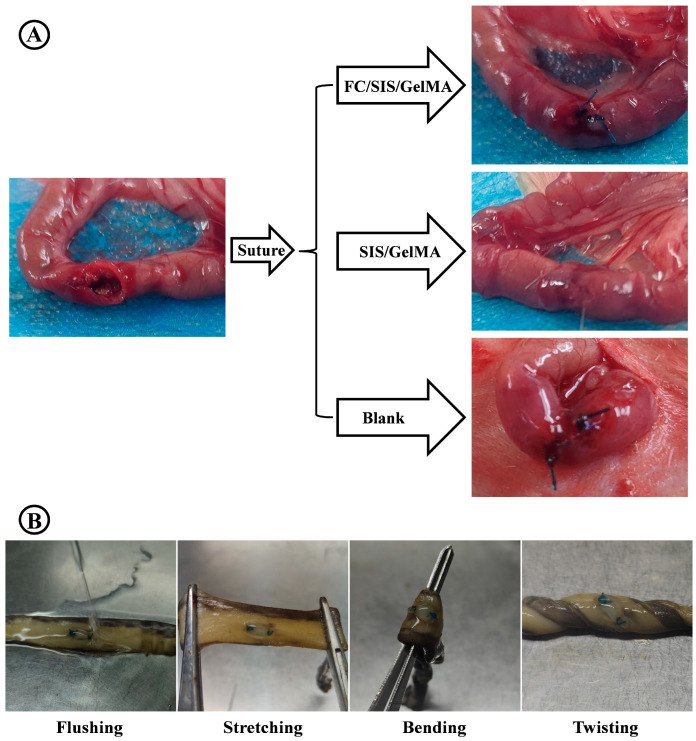
The intestinal defect repair experiment. (**A**) The schematic shows the experimental procedure for creating intestinal defects using the SD rat animal model. (**B**) FC/SIS/GelMA hydrogel maintained stable adhesion and integrity on the small intestinal tissue crack site after stretching, twisting, bending, and water flushing.

**Figure 10 ijms-26-00663-f010:**
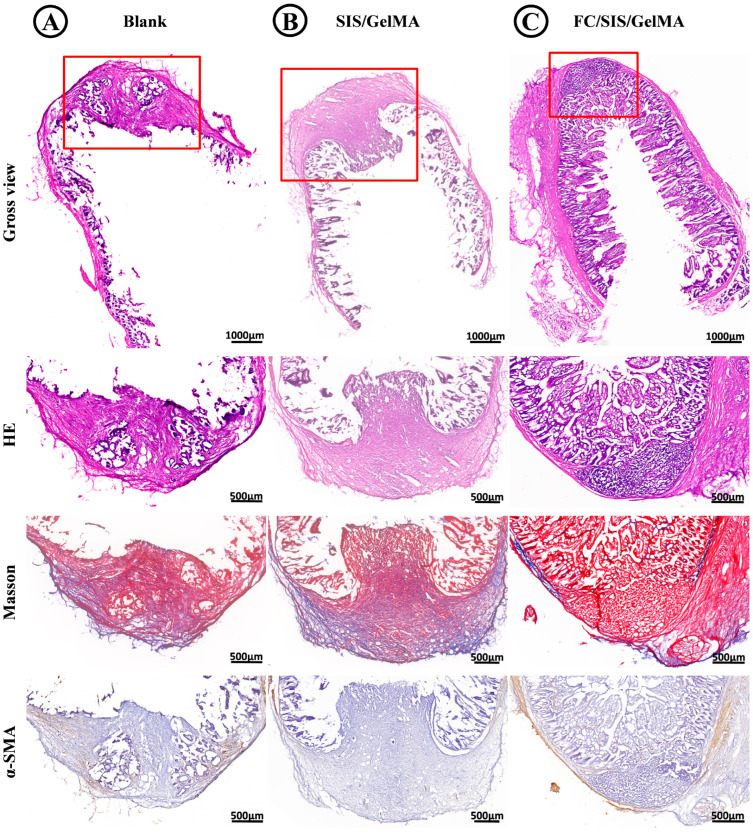
Therapeutic results of intestinal defect repair. (**A**) Histological examinations of H&E, Masson, and α-SMA staining in blank group. (**B**) Histological examinations of H&E, Masson, and α-SMA staining in SIS/GelMA group. (**C**) Histological examinations of H&E, Masson, and α-SMA staining in FC/SIS/GelMA group. Red boxes, Enlarged section.

**Table 1 ijms-26-00663-t001:** Final concentrations of hydrogels.

	Fish Collagen (mg/mL)	Small Intestinal Submucosa (mg/mL)	GelMA (mg/mL)
1	5	10	50
2	10	10	50
3	15	10	50

**Table 2 ijms-26-00663-t002:** Primer sequences.

Primer Name	Forward	Reverse
VILLIN 1	ATGACTCCAGCTGCCTTCTCT	GCTCTGGGTTAGAGCTGTAAG
LGR5	ACCCGCCAGTCTCCTACATC	GCATCTAGGCGCAGGGATTG
MUC2	ATGCCCACCTCCTCAAAGAC	GTAGTTTCCGTTGGAACAGTGAA
OLFM4	CAGCCACTTTCCAATTTCACTG	GCTGGACATACTCCTTCACCTTA
CHGA	CTCGTCCACTCTTTCCGCAC	CTGGGTTTGGACAGCGAGTC
CLDN18	ACATGCTGGTGACTAACTTCTG	AAATGTGTACCTGGTCTGAACAG
KRT20	TTCAGTCGTCAAAGTTTTCACCG	TCCTATACAGCGAGCCACTCA
LYZ	GAGACCGAAGCACCGACTATG	CGGTTTTGACATTGTGTTCGC

## Data Availability

The data presented in this study are available on request from the corresponding author. The data are not publicly available due to: follow-up trials are still ongoing.

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
