# Peer review of "Photo-Crosslinking Hydrogel Based on Porcine Small Intestinal Submucosa Decellularized Matrix/Fish Collagen/GelMA for Culturing Small Intestinal Organoids and Repairing Intestinal Defects"

_ijms, 2025, doi:10.3390/ijms26020663_

Round 1
Reviewer 1 Report
Comments and Suggestions for Authors
Regarding the manuscript ID: ijms-3397125
Type: Article
Entitled: Photo-crosslinking hydrogel based on porcine small intestinal submucosa decellularized matrix/fish collagen/GelMA for culturing small intestinal organoids and repairing intestinal defects
The manuscript exhibits a well-designed layout, its contents are user-friendly, and it clearly states the study's objectives and rationale. The study showcases an outstanding application of biomaterials hydrogels in the field of medicine. The data supports the interpretation of results and study conclusions, and the topic aligns with the journal's scope. However, some improvements are necessary to make it suitable for publication. Based on these points, I would like to recommend publishing after revision. Some comments are below:
- Some parts of the introduction and discussion lack relevant references; please revise them with updated references.
- Some parts of the methodssuch as “swelling, degradation and porosity” and discussion lack relevant references; please revise them with updated references from 2024 and 2023.
- The English expression in the manuscript needs further improvement.
- Why do most references in the bibliography section include the word “etal” and not complete the authors description? For example, in reference 31,you write “Xu, Z.Y., et al., Extracellular matrix bioink boosts stemness and facilitates transplantation of intestinal organoids as a biosafe Matrigel alternative. Bioeng Transl Med, 2023. 8(1): p. e10327” which should be as follows: “Xu, Z.Y.; Huang, J.J.; Liu, Y.; Chen, C.W.; Qu, G.W.; Wang, G.F.; Zhao, Y.; Wu, X.W.; Ren, J.A. Extracellular matrix bioink boosts stemness and facilitates transplantation of intestinal organoids as a biosafe Matrigel alternative. Bioeng Transl Med2023, 8, p. e10327.” to be compatible with the journal scope.
- In the section on the swelling experiment, you mentioned that you soaked the samples for 30 minutes. Could you explain why? Did you study swelling kinetics against time to reach equilibrium or plateau, or do you have a reference confirming soaking for 30 minutes?
- In the hydrogel preparation section, the curing time is 30 seconds, compared to approximately 900 seconds for Matrigel. This is an excellent result. However, why is the curing time only 30 seconds, even though reference 31 did not provide any details? Did you conduct a solubility and gel fraction study to confirm that 30 seconds is the optimal duration for your formula's gelation?
- In the mechanical properties discussion according to Fig. 2D, I think maybe the stress values are very high, but the strain values are small; this may mean the hydrogel is tough, not elastic. Is this better to mimic ECM structure? Could you improve this section by having a more thorough discussion?
- the English expression in the manuscript needs further improvement.
Author Response
Dear Reviewer,
Thank you very much for your valuable comments on our research, your comments are very helpful to the rigor, comprehensiveness, scientificity, and standardization of our article, the following is our response to you:
Comments 1: Some parts of the introduction and discussion lack relevant references; please revise them with updated references.
Responds 1: Thank you very much for your valuable comments. We have added some references in the introduction and discussion section. The modifications are highlighted in red in the manuscript.
Comments 2: Some parts of the methods such as “swelling, degradation and porosity” and discussion lack relevant references; please revise them with updated references from 2024 and 2023.
Responds 2: Thank you very much for your valuable comments, we have added references in the corresponding positions according to your comments, and the changed parts are marked with a red mark in the manuscript.
Comments 3: The English expression in the manuscript needs further improvement.
Responds 3: Thank you very much for your valuable comments. We have asked a professional language polishing agency to comprehensively revise our article.
Comments 4: Why do most references in the bibliography section include the word “etal” and not complete the authors description? For example, in reference 31,you write “Xu, Z.Y., et al., Extracellular matrix bioink boosts stemness and facilitates transplantation of intestinal organoids as a biosafe Matrigel alternative. Bioeng Transl Med, 2023. 8(1): p. e10327” which should be as follows: “Xu, Z.Y.; Huang, J.J.; Liu, Y.; Chen, C.W.; Qu, G.W.; Wang, G.F.; Zhao, Y.; Wu, X.W.; Ren, J.A. Extracellular matrix bioink boosts stemness and facilitates transplantation of intestinal organoids as a biosafe Matrigel alternative. Bioeng Transl Med2023, 8, p. e10327.” to be compatible with the journal scope.
Responds 4: Thank you very much for your valuable suggestions, and we have revised the format of the references according to your requirements.
Comments 5: In the section on the swelling experiment, you mentioned that you soaked the samples for 30 minutes. Could you explain why? Did you study swelling kinetics against time to reach equilibrium or plateau, or do you have a reference confirming soaking for 30 minutes?
Responds 5: Thank you very much for your valuable advice. We used the experimental steps in the following article: Wang, D., Guo, Y., Zhu, J., Liu, F., Xue, Y., Huang, Y., Zhu, B., Wu, D., Pan, H., Gong, T., Lu, Y., Yang, Y., & Wang, Z. (2023). Hyaluronic acid methacrylate/pancreatic extracellular matrix as a potential 3D printing bioink for constructing islet organoids. Acta biomaterialia, 165, 86–101. https://doi.org/10.1016/j.actbio.2022.06.036. According to your suggestion, we reviewed the relevant articles again and found that 30 minutes of immersion is not very rigorous [1-3]. Therefore, we extended the immersion time to 1 day according to the relevant articles and re-examined the test, and the modifications (Figure2B) are marked with red marks in the manuscript. The references are as follows:
- Wang Y, Ma M, Wang J, et al. Development of a Photo-Crosslinking, Biodegradable GelMA/PEGDA Hydrogel for Guided Bone Regeneration Materials. Materials (Basel). 2018;11(8):1345. Published 2018 Aug 3. doi:10.3390/ma11081345
- Camci-Unal G, Cuttica D, Annabi N, Demarchi D, Khademhosseini A. Synthesis and characterization of hybrid hyaluronic acid-gelatin hydrogels. Biomacromolecules. 2013;14(4):1085-1092. doi:10.1021/bm3019856
- Qi H, Wang B, Wang M, Xie H, Chen C. A pH/ROS-responsive antioxidative and antimicrobial GelMA hydrogel for on-demand drug delivery and enhanced osteogenic differentiation in vitro. Int J Pharm. 2024;657:124134. doi:10.1016/j.ijpharm.2024.124134
Finally, thank you once again for your valuable suggestions, which will help us tremendously in the rigor and science of our article!
Comments 6: In the hydrogel preparation section, the curing time is 30 seconds, compared to approximately 900 seconds for Matrigel. This is an excellent result. However, why is the curing time only 30 seconds, even though reference 31 did not provide any details? Did you conduct a solubility and gel fraction study to confirm that 30 seconds is the optimal duration for your formula's gelation?
Responds 6: Thank you very much for your valuable comments which pointed out the shortcomings of our study. We hope to significantly reduce the cross-linking time by using GelMA as a carrier, here is the instruction manual of the GelMA hydrogel we purchased. We are very sorry for being limited by the conditions of the laboratory, we only judged the general gelation time by observing and turning the petri dish, and we really could not accurately determine the optimal gelation time. In addition, most of the components in the hydrogel are GelMA, so we believe that the GelMA gelation time can reflect the gelation time of the hydrogel to a certain extent. In the future, we will look for suitable tests to determine the optimal gelation time based on your comments, hope you can understand.

Translation of text marked in red: 405nm light source, irradiation for 10-30 seconds to make gelation, can be adjusted through the light time and intensity of the gel strength.
Comments 7: In the mechanical properties discussion according to Fig. 2D, I think maybe the stress values are very high, but the strain values are small; this may mean the hydrogel is tough, not elastic. Is this better to mimic ECM structure? Could you improve this section by having a more thorough discussion?
Responds 7: Thank you very much for your comments, this is indeed a shortcoming of our experiment. The aim of this study was to investigate whether hydrogels could be used to culture small intestinal organoids. Therefore, we just chose the relatively most suitable formulation among the three groups of hydrogels we designed, but it is really not the most ideal formulation, therefore, we modified the following in the discussion section. It is highlighted in red in the manuscript(Paragraph 14 of the discussion).
While the hydrogel mimicked the structure of ECM to some extent, its mechanical properties were greater than those of th intestine. Excessive mechanical strength hindered the growth of small intestinal organoids, whereas insufficient mechanical strength compromised intestinal repair. Therefore, future experiments, should focus on adjusting the concentration ratios of the three components to determine the optimal concentration of the hydrogel formulation.
Thank you again for your comments, which have significantly improved the scientific, comprehensive and rigorous nature of this article.
Thank you for your valuable advice and best regards. If you have any other comments or suggestions, we welcome your further guidance.
Yours sincerely,
Ziwei Wang
(Please see the attachment)

Reviewer 2 Report
Comments and Suggestions for Authors
This manuscript presents an innovative photo-crosslinkable hydrogel (FC/SIS/GelMA) designed as a cost-effective, biocompatible, and clinically relevant alternative to Matrigel for culturing small intestinal organoids and repairing intestinal defects. The authors comprehensively characterize the hydrogel’s properties and demonstrate its capacity to support organoid growth and promote tissue regeneration both in vitro and in vivo. The work is novel, clinically relevant, and methodologically robust, addressing significant limitations of existing biomaterials. The following issues need to be addressed before publication:
(1) A more detailed discussion of other alternative hydrogels (e.g., alginate, PEG-4MAL) and their limitations would provide additional context and strengthen the argument for the novelty of the presented material.
(2) Details about the tests performed (e.g., ANOVA, t-tests) and the justification for sample sizes are missing. Including these details will enhance the rigor and reproducibility of the findings.
(3) The clinical implications of FC/SIS/GelMA hydrogel for tissue engineering and regenerative medicine are significant. Including a discussion of how this hydrogel could be adapted for other organoid systems or tissue types would broaden its relevance and appeal.
(4) The manuscript could benefit from further discussion on the observed reduction in organoid diameter when using FC/SIS/GelMA compared to Matrigel. Speculating on the underlying causes (e.g., enhanced mechanical stiffness) and potential solutions (e.g., optimizing FC concentration) would improve the clarity and impact of the discussion.
Author Response
Dear Reviewer,
Thank you very much for your valuable comments, your comments have significantly improved the accuracy, standardization, comprehensiveness and systematization of our articles, we have been revised in accordance with your comments, the following is a reply to you:
Comments 1: A more detailed discussion of other alternative hydrogels (e.g., alginate, PEG-4MAL) and their limitations would provide additional context and strengthen the argument for the novelty of the presented material.
Responds 1:
Thank you very much for your valuable comments, which have helped us tremendously in making our article comprehensive and rigorous. Based on your comments, we have added the following to Paragraph 3 of the discussion section.
However, the aforementioned studies exhibited certain limitations. The success rate of intestinal organoids cultured in alginate was significantly lower than that of Matrigel, and the mortality rate of the organoids at the later stage of culture (after 28 days) was also significantly higher than that of Matrigel. Lower organoid yields in alginate as compared to Matrigel may be due to the inability of cells to remodel the hydrogel, lack of interactions with serum proteins, or the lack of growth factors present in Matrigel. Regarding collagen-nanocellulose hydrogels, dissected crypts directly seeded in collagen-nanocellulose hydrogel do not progress into organoids. Only when a small amount of Matrigel-20% (v/v) is added to the hydrogel, fresh intestinal crypts will form organoids. It could also be a consequence of the lack of certain growth factors in the hydrogels. As for PEG-4MAL hydrogels, an important aspect of this engineered hydrogel platform is its rapid reaction kinetics, which, if mixing is not conducted properly, can lead to the formation of an inhomogeneous gel that presents variability in its physicochemical properties. This seriously limits the feasibility of its clinical application.
Comments 2: Details about the tests performed (e.g., ANOVA, t-tests) and the justification for sample sizes are missing. Including these details will enhance the rigor and reproducibility of the findings.
Responds 2:
Thank you very much for your comments. We have revised the following contents in the Statistical Analysis section according to your comments, which are marked in red in the manuscript.
Statistical analyses and graphical representations of the data from this study were mostly generated using GraphPad Prism 8(GraphPad Software, La Jolla, CA, USA). Results are presented as mean ± S.D. The unpaired, two-sided Student’s t-tests with 95% and 99% confidence intervals was used to determine the significance of the data between the two groups. One-way analysis of variance was conducted to determine the significance of data with more than two groups and was followed by Tukey’s multiple comparisons test. Throughout the study, the sample size was determined based on our preliminary studies and on the criteria in the field. At least three biological samples were included for one experiment and one to three independent experiments were performed to ensure sufficient reproducibility of the results.
Comments 3: The clinical implications of FC/SIS/GelMA hydrogel for tissue engineering and regenerative medicine are significant. Including a discussion of how this hydrogel could be adapted for other organoid systems or tissue types would broaden its relevance and appeal.
Responds 3:
Thank you very much for your valuable comments. We have added relevant content in the penultimate paragraph of the discussion section, and the modified content is marked with red marks. The specific changes are as follows.
For example, Satoshi Watanabe et al. infused colonic organoid tissue into the luminal space via the anus. The infused organoids subsequently attach to the injured region and rebuild a donor-derived epithelium. This study provides an experimental basis for the treatment of refractory ulcerative colitis using colonoid organ transplantation therapy. If FC/SIS/GelMA hydrogel can be co-cultured with colonic organoids in vitro and then injected into the damaged intestinal tract, in situ cross-linking will enable it to be firmly affixed to the surface of the damaged intestinal tract. This will bring the following benefits, firstly, the FC/SIS/GelMA hydrogel has the effect of promoting the repair of intestinal defects, and secondly, it can make the organoids tightly bond to the damaged site, providing an ideal carrier for the organoids. This will significantly improve the therapeutic effect of colitis.
Comments 4: The manuscript could benefit from further discussion on the observed reduction in organoid diameter when using FC/SIS/GelMA compared to Matrigel. Speculating on the underlying causes (e.g., enhanced mechanical stiffness) and potential solutions (e.g., optimizing FC concentration) would improve the clarity and impact of the discussion.
Responds 4:
Thank you very much for your valuable comments, which have significantly improved the rigor and comprehensiveness of this paper. Based on your comments, we have added the following to the discussion section. ( third last paragraph)
Last, Compared with Matrigel, the diameter of the cultured organoids in FC/SIS/GelMA hydrogel was relatively small, and analyzing the reasons, we believe that the main reason is that the mechanical properties of FC/SIS/GelMA hydrogel are higher than that of Matrigel, and a larger mechanical strength is not conducive to the growth of the organoids, which restricts the diameter of the organoids. But a too-low mechanical strength is not conducive to the control of the morphology of the hydrogel as well as to the promotion of the gut defect repair, therefore, in subsequent experiments, optimizing the concentration ratio of the three components and exploring the most ideal hydrogel concentration ratio remains essential.
We appreciate your review work and your comments are very useful for our research and significantly improve the professionalism of our article. If you have any other comments or suggestions, we welcome your further guidance.
Thank you again for your help and support!
Yours sincerely,
Ziwei Wang
Please see the attachment
